# Modeling Contextual Relationships Among Utterances for Multimodal Sentiment Analysis

## Abstract

Multimodal sentiment analysis is a developing area of research, which involves identification of emotions and sentiments in videos. Current research considers utterances as independent entities, i.e., ignores the inter-dependencies and relations among utterances of a video. In this paper, we propose an LSTM based model which enables these utterances to capture contextual information from its surroundings in the same video, thus aiding the classification process. Our model shows $5-10\%$ improvement over the state of the art and high robustness to generalizability.

## 1 Introduction

Emotion recognition and sentiment analysis have become a new trend in social media, helping users to automatically extract the opinions expressed in user-generated content, especially videos. Thanks to the high availability of computers and smartphones, and the rapid rise of social media, consumers tend to record their reviews and opinions about products or films and upload them on social media platforms, such as YouTube or Facebook. Such videos often contain comparisons, which can aid prospective buyers make an informed decision.

The primary advantage of analyzing videos over text is the surplus of behavioral cues present in vocal and visual modalities. The vocal modulations and facial expressions in the visual data, along with textual data, provide important cues to better identify affective states of the opinion holder. Thus, a combination of text and video data helps to create a better emotion and sentiment analysis model (Poria et al., 2017).

Recently, a number of approaches to multimodal sentiment analysis, producing interesting results, have been proposed (Pérez-Rosas et al., 2013; Wollmer et al., 2013; Poria et al., 2015). However, there are major issues that remain unaddressed, such as the role of speaker-dependent versus speaker-independent models, the impact of each modality across the dataset, and generalization ability of a multimodal sentiment classifier. Leaving these issues unaddressed has presented difficulties in effective comparison of different multimodal sentiment analysis methods.

An utterance is a unit of speech bound by breathes or pauses. Utterance-level sentiment analysis focuses on tagging every utterance of a video with a sentiment label (instead of assigning a unique label to the whole video). In particular, utterance-level sentiment analysis is useful to understand the sentiment dynamics of different aspects of the topics covered by the speaker throughout his/her speech. The true meaning of an utterance is relative to its surrounding utterances.

In this paper, we consider such surrounding utterances to be the context, as the consideration of temporal relation and dependency among utterances is key in human-human communication. For example, the MOSI dataset (Zadeh et al., 2016) contains a video, in which a girl reviews the movie 'Green Hornet'. At one point, she says "The Green Hornet did something similar". Normally, doing something similar, i.e., monotonous or repetitive might be perceived as negative. However, the nearby utterances "It engages the audience more", "they took a new spin on it", "and I just loved it" indicate a positive context.

In this paper, we discard the oversimplifying hypothesis on the independence of utterances and develop a framework based on long short-term memory (LSTM) to extract utterance features that also consider surrounding utterances.

Our model enables consecutive utterances to share information, thus providing contextual information in the classification process. Experimental results show that the proposed framework has outperformed the state of the art on benchmark datasets by $5-10\%$. The paper is organized as follows: Section 2 provides a brief literature review on multimodal sentiment analysis; Section 3 describes the proposed method in detail; experimental results and discussion are shown in Section 4; finally, Section 5 concludes the paper.

## 2 Related Work

Text-based sentiment analysis systems can be broadly categorized into knowledge-based and statistics-based systems (Cambria, 2016). While the use of knowledge bases was initially more popular for the identification of emotions and polarity in text, sentiment analysis researchers have recently been using statistics-based approaches, with a special focus on supervised statistical methods (Pang et al., 2002; Socher et al., 2013).

In 1970, Ekman (Ekman, 1974) carried out extensive studies on facial expressions which showed that universal facial expressions are able to provide sufficient clues to detect emotions. Recent studies on speech-based emotion analysis (Datcu and Rothkrantz, 2008) have focused on identifying relevant acoustic features, such as fundamental frequency (pitch), intensity of utterance, bandwidth, and duration.

As for fusing audio and visual modalities for emotion recognition, two of the early works were done by De Silva et al. (De Silva et al., 1997) and Chen et al. (Chen et al., 1998). Both works showed that a bimodal system yielded a higher accuracy than any unimodal system. More recent research on audio-visual fusion for emotion recognition has been conducted at either feature level (Kessous et al., 2010) or decision level (Schuller, 2011).

While there are many research papers on audio-visual fusion for emotion recognition, only a few have been devoted to multimodal emotion or sentiment analysis using textual clues along with visual and audio modalities. Wollmer et al. (Wollmer et al., 2013) and Rozgic et al. (Rozgic et al., 2012a,b) fused information from audio, visual, and textual modalities to extract emotion and sentiment. Metallinou et al. (Metallinou et al., 2008) and Eyben et al. (Eyben et al., 2010a) fused audio and textual modalities for emotion recognition.

Both approaches relied on a feature-level fusion. Wu et al. (Wu and Liang, 2011) fused audio and textual clues at decision level.

## 3 Method

In this work, we propose a LSTM network that takes as input all utterances in a video and extracts contextual unimodal and multimodal features by modeling the dependencies among the input utterances. Below, we propose an overview of the method -

**1. Context-Independent Unimodal Utterance-Level Feature Extraction**

First, the unimodal features are extracted without considering the contextual information of the utterances (Section 3.1). Table 1 presents the feature extraction methods used for each modality.

**2. Contextual Unimodal and Multimodal Classification**

The context-independent unimodal features (from Step 1) are then fed into a LSTM network (termed contextual LSTM) that allows consecutive utterances in a video to share semantic information in the feature extraction process (which provides context-dependent unimodal and multimodal classification of the utterances). We experimentally show that this proposed framework improves the performance of utterance-level sentiment classification over traditional frameworks.

Videos, comprising of its constituent utterances, serve as the input. We represent the dataset as U:

$$U = \begin{bmatrix} u_{1,1} & u_{1,2} & u_{1,3} & ... & u_{1,L_1} \\ u_{2,1} & u_{2,2} & u_{2,3} & ... & u_{2,L_2} \\ . & . & . & ... & . \\ u_{M,1} & u_{M,2} & u_{M,3} & ... & u_{M,L_M} \end{bmatrix}.$$

Here, $u_{i,j}$ denotes the $j^{th}$ utterance of the $i^{th}$ video and $L = [L_1, L_2, ..., L_M]$ represents the number of utterances per video in the dataset set.

### 3.1 Extracting Context-Independent Unimodal Features

Initially, the unimodal features are extracted from each utterance separately, i.e., we do not consider the contextual relation and dependency among the utterances (Table 1). Below, we explain the textual, audio, and visual feature extraction methods.

### 3.1.1 text-CNN: Textual Features Extraction

For feature extraction from textual data, we use a convolutional neural network (CNN).

The idea behind convolution is to take the dot product of a vector of $k$ weights, $w_k$, known as kernel vector, with each $k$-gram in the sentence $s(t)$ to obtain another sequence of features $c(t) = (c_1(t), c_2(t), \ldots, c_L(t))$:

$$c_j = w_k^T \cdot \mathbf{x}_{i:i+k-1}.$$

We then apply a max pooling operation over the feature map and take the maximum value $\hat{c}(t) = \max\{\mathbf{c}(t)\}$ as the feature corresponding to this particular kernel vector. We use varying kernel vectors and window sizes to obtain multiple features.

The process of extracting textual features is as follows -

First, we represent each sentence as the concatenation of vectors of the constituent words. These vectors are the publicly available 300-dimensional word2vec vectors trained on 100 billion words from Google News (Mikolov et al., 2013). The convolution kernels are thus applied to these word vectors instead of individual words. Each sentence is wrapped to a window of 50 words which serves as the input to the CNN.

The CNN has two convolutional layers - the first layer having a kernel size of 3 and 4, with 50 feature maps each and a kernel size 2 with 100 feature maps for the second. The convolution layers are interleaved with pooling layers of dimension 2. We use ReLU as the activation function. The convolution of the CNN over the sentence learns abstract representations of the phrases equipped with implicit semantic information, which with each successive layer spans over increasing number of words and ultimately the entire sentence.

| Modality | Model |
|----------|-------|
| Text | *text-CNN*: Deep Convolutional Neural Network with word embeddings |
| Video | *3d-CNN*: 3-dimensional CNNs employed on utterances of the videos |
| Audio | *openSMILE*: Extracts low level audio descriptors from the audio modality |

Table 1: Methods for extracting context independent baseline features from different modalities.

### 3.1.2 Audio Feature Extraction

Audio features are extracted in 30 Hz frame-rate; we use a sliding window of 100 ms. To compute the features, we use the open-source software openSMILE (Eyben et al., 2010b) which automatically extracts pitch and voice intensity. Voice normalization is performed and voice intensity is thresholded to identify samples with and without voice. Z-standardization is used to perform voice normalization.

The features extracted by openSMILE consist of several low-level descriptors (LLD) and their statistical functionals. Some of the functionals are amplitude mean, arithmetic mean, root quadratic mean, etc. Taking into account all functionals of each LLD, we obtained 6373 features.

### 3.1.3 Visual Feature Extraction

We use 3D-CNN to obtain visual features from the video. We hypothesize that 3D-CNN will not only be able to learn relevant features from each frame, but will also be able to learn the changes among given number of consecutive frames.

In the past, 3D-CNN has been successfully applied to object classification on 3D data (Ji et al., 2013). Its ability to achieve state-of-the-art results motivated us to use it.

Let $vid \in \mathbb{R}^{c \times f \times h \times w}$ be a video, where $c$ = number of channels in an image (in our case $c = 3$, since we consider only RGB images), $f$ = number of frames, $h$ = height of the frames, and $w$ = width of the frames. Again, we consider the 3D convolutional filter $filt \in \mathbb{R}^{fm \times c \times fl \times fh \times fw}$, where $fm$ = number of feature maps, $c$ = number of channels, $fd$ = number of frames (in other words depth of the filter), $fh$ = height of the filter, and $fw$ = width of the filter. Similar to 2D-CNN, $filt$ slides across video $vid$ and generates output $convout \in \mathbb{R}^{fm \times c \times (f-fd+1) \times (h-fh+1) \times (w-fw+1)}$. Next, we apply max pooling to $convout$ to select only relevant features. The pooling will be applied only to the last three dimensions of the array $convout$.

In our experiments, we obtained best results with 32 feature maps ($fm$) with the filter-size of $5 \times 5 \times 5$ (or $fd \times fh \times fw$). In other words, the dimension of the filter is $32 \times 3 \times 5 \times 5 \times 5$ (or $fm \times c \times fd \times fh \times fw$). Subsequently, we apply max pooling on the output of convolution operation, with window-size being $3 \times 3 \times 3$. This is followed by a dense layer of size 300 and softmax. The activations of this dense layer are finally used as the video features for each utterance.

### 3.2 Context-Dependent Feature Extraction

We hypothesize that, within a video, there is a high probability of utterance relatedness with respect

to their sentimental and emotional clues. Since most videos tend to be about a single topic, the utterances within each video are correlated, e.g., due to the development of the speaker's idea, co-references, etc. This calls for a model which takes into account such inter-dependencies and the effect these might have on the current utterance. To capture this flow of informational triggers across utterances, we use a LSTM-based recurrent network scheme (Gers, 2001).

### 3.2.1 Long Short-Term Memory

LSTM is a kind of recurrent neural network (RNN), an extension of conventional feed-forward neural network. Specifically, LSTM cells are capable of modeling long-range dependencies, which other traditional RNNs fail to do given the vanishing gradient issue. Each LSTM cell consists of an input gate $i$, an output gate $o$, and a forget gate $f$, which enables it to remember the error during the error propagation. Current research (Zhou et al., 2016) indicates the benefit of using such networks to incorporate contextual information in the classification process.

In our case, the LSTM network serves the purpose of context-dependent feature extraction by modeling relations among utterances. We term our architecture 'contextual LSTM'. We propose several architectural variants of it later in the paper.

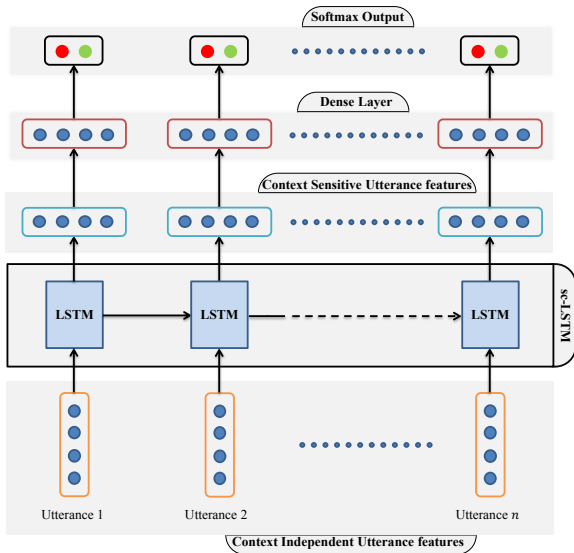

Figure 1: Contextual LSTM network: input features are passed through an unidirectional LSTM layer, followed by a dense layer and then a softmax layer. Categorical cross entropy loss is taken for training. The dense layer activations serve as the output features.

### 3.2.2 Contextual LSTM Architecture

Let unimodal features have dimension $k$, each utterance is thus represented by a feature vector $\boldsymbol{x}_{i,t} \in \mathbb{R}^k$, where $t$ represents the $t^{th}$ utterance of the video $i$. For a video, we collect the vectors for all the utterances in it, to get $\boldsymbol{X_i} = [\boldsymbol{x}_{i,1}, \boldsymbol{x}_{i,2}, ..., \boldsymbol{x}_{i,L_i}] \in \mathbb{R}^{L_i \times k}$, where $L_i$ represents the number of utterances in the video. This matrix $X_i$ serves as the input to the LSTM. Figure 1 demonstrates the functioning of this LSTM module.

In the procedure *getLstmFeatures($X_i$)* of Algorithm 1, each of these utterance $x_{i,t}$ is passed through a LSTM cell using the equations mentioned in line 32 to 37. The output of the LSTM cell $h_{i,t}$ is then fed into a dense layer and finally into a softmax layer (line 38 to 39). The activations of the dense layer $z_{i,t}$ are used as the context-dependent features of contextual LSTM.

### 3.2.3 Training

The training of the LSTM network is performed using categorical cross entropy on each utterance's softmax output per video, i.e.,

$$loss = \frac{1}{N} \sum_{n=1}^{N} \sum_{c=1}^{C} y_{n,c} \log_2(\hat{y_{n,c}}),$$

where $N$ = total number of utterances in a video, $y_{n,c}$ = original output of class $c$, and $\hat{y_{n,c}}$ = predicted output.

A dropout layer between the LSTM cell and dense layer is introduced to check overfitting. As the videos do not have same the number of utterances, padding is introduced to serve as neutral utterances. To avoid the proliferation of noise within the network, masking is done on these padded utterances to eliminate their effect in the network. Parameter tuning is done on the train set by splitting it into train and validation components with 80/20% split. RMSprop has been used as the optimizer which is known to resolve Adagrad's radically diminishing learning rates (Duchi et al., 2011). After feeding the train set to the network, the test set is passed through it to generate their context-dependent features.

**Different Network Architectures** We consider the following variants of the contextual LSTM architecture in our experiments -

**sc-LSTM** This variant of the contextual LSTM architecture consists of unidirectional

LSTM cells. As this is the simple variant of the contextual LSTM, we termed it as simple contextual LSTM (sc-LSTM)

**h-LSTM** We also test on an architecture where the dense layer after the LSTM cell is omitted. Thus, the output of the LSTM cell $h_{i,t}$ provides our context-dependent features and the softmax layer provides the classification. We call this architecture hidden LSTM (h-LSTM).

**bc-LSTM** Bi-directional LSTMs are two unidirectional LSTMs stacked together having opposite directions. Thus, an utterance can get information from other utterances occurring before and after itself in the video. We replaced the regular LSTM with a bi-directional LSTM and named the resulting architecture as bi-directional contextual LSTM (bc-LSTM). The training process of this architecture is similar to sc-LSTM.

**uni-SVM** In this setting, we first obtain the unimodal features as explained in Section 3.1, concatenate them and then send to a SVM for the final classification. It should be noted that using a gated recurrent unit (GRU) instead of LSTM did not improve the performance.

### 3.3   Fusion of Modalities

We accomplish multimodal fusion in two different ways as explained below -

#### 3.3.1   Non-hierarchical Framework

In non-hierarchical framework, we concatenate context-independent unimodal features (from Section 3.1) and feed that into the contextual LSTM networks, i.e., sc-LSTM, bc-LSTM, and h-LSTM.

#### 3.3.2   Hierarchical Framework

Contextual unimodal features, taken as input, can further improve performance of the multimodal fusion framework explained in Section 3.3.1. To accomplish this, we propose a hierarchical deep network which comprises of two levels –

**Level-1:** context-independent unimodal features (from 3.1) are fed to the proposed LSTM network (Section 3.2.2) to get *context-sensitive* unimodal feature representations for each utterance. Individual LSTM networks are used for each modality.

**Level-2:** consists of a contextual LSTM network similar to Level-1 but independent in training and computation. Output from each LSTM network in Level-1 are concatenated and fed into

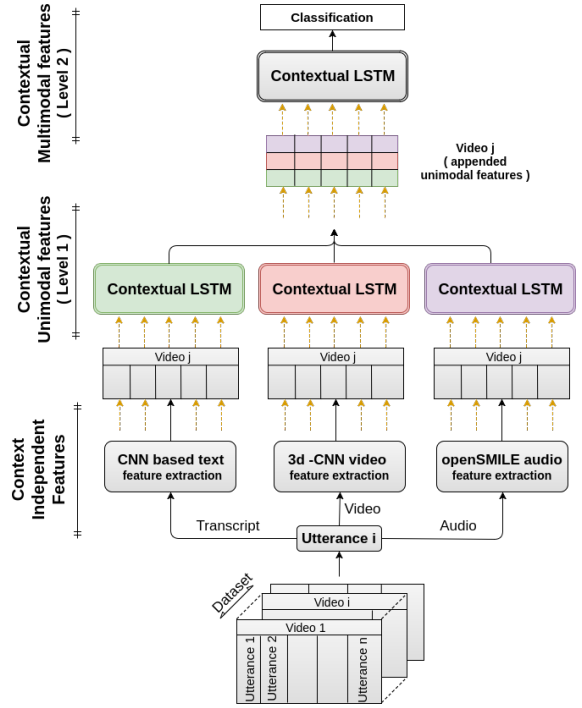

Figure 2: Hierarchical architecture for extracting context-dependent multimodal utterance features. LSTM module has been described in Figure 1.

this LSTM network, thus providing an inherent fusion scheme - the prime objective of this level (Fig 2). The performance of the second level banks on the quality of the features from the previous level, with better features aiding the fusion process. Algorithm 1 describes the overall computation for utterance classification. For the hierarchical framework, we train Level 1 and Level 2 successively but separately.

| Weight | | Bias | |
|---|---|---|---|
| $W_i, W_f, W_c, W_o$ | $\in \mathbb{R}^{d \times k}$ | $b_i, b_f, b_c, b_o$ | $\in \mathbb{R}^d$ |
| $P_i, P_f, P_c, P_o V_o$ | $\in \mathbb{R}^{d \times d}$ | $b_z$ | $\in \mathbb{R}^m$ |
| $W_z$ | $\in \mathbb{R}^{m \times d}$ | $b_{sft}$ | $\in \mathbb{R}^c$ |
| $W_{sft}$ | $\in \mathbb{R}^{c \times m}$ | | |

Table 2: Summary of notations used in Algorithm 1. Note: $d$ - dimension of hidden unit. $k$ - dimension of input vectors to LSTM layer . $c$ - number of classes.

## 4   Experimental Results

### 4.1   Dataset details

Most of the research in multimodal sentiment analysis is performed on datasets with speaker overlap in train and test splits.

Because each individual has a unique way of expressing emotions and sentiments, finding generic, person-independent features for sentimental analysis is very tricky. In real-world applications, the

**Algorithm 1** Proposed Architecture

---

1: **procedure** TRAINARCHITECTURE( U, V)
2: Train context-independent models with $U$
3: **for** i:[1,M] **do** ▷ extract baseline features
4: **for** j:[1,$L_i$] **do**
5: $x_{i,j} \leftarrow TextFeatures(u_{i,j})$
6: $x'_{i,j} \leftarrow VideoFeatures(u_{i,j})$
7: $x''_{i,j} \leftarrow AudioFeatures(u_{i,j})$

8: **Unimodal**:
9: Train LSTM at Level-1 with $X, X' and X''$.
10: **for** i:[1,M] **do** ▷ unimodal features
11: $Z_i \leftarrow getLSTMFeatures(X_i)$
12: $Z'_i \leftarrow getLSTMFeatures(X'_i)$
13: $Z''_i \leftarrow getLSTMFeatures(X''_i)$

14: **Multimodal**:
15: **for** i:[1,M] **do**
16: **for** j:[1,$L_i$] **do**
17: **if** Non-hierarchical fusion **then**
18: $x^*_{i,j} \leftarrow (x_{i,j}\|x'_{i,j}\|x''_{i,j})$ ▷ concatenation
19: **else**
20: **if** Hierarchical fusion **then**
21: $x^*_{i,j} \leftarrow (z_{i,j}\|z'_{i,j}\|z''_{i,j})$ ▷ concatenation
22: Train LSTM at Level-2 with $X^*$.
23: **for** i:[1,M] **do** ▷ multimodal features
24: $Z^*_i \leftarrow getLSTMFeatures(X^*_i)$
25: testArchitecture( V)
26: **return** $Z^*$

27: **procedure** TESTARCHITECTURE( V)
28: Similar to training phase. V is passed through the learnt models to get the features and classification outputs. Table 2 shows the trainable parameters.

29: **procedure** GETLSTMFEATURES($X_i$) ▷ for $i^{th}$ video
30: $Z_i \leftarrow \phi$
31: **for** t:[1,$L_i$] **do** ▷ Table 2 provides notation
32: $i_t \leftarrow \sigma(W_i x_{i,t} + P_i.h_{t-1} + b_i)$
33: $\widetilde{C_t} \leftarrow tanh(W_c x_{i,t} + P_c h_{t-1} + b_c)$
34: $f_t \leftarrow \sigma(W_f x_t + P_f h_{t-1} + b_f)$
35: $C_t \leftarrow i_t * \widetilde{C_t} + f_t * C_{t-1}$
36: $o_t \leftarrow \sigma(W_o x_t + P_o h_{t-1} + V_o C_t + b_o)$
37: $h_t \leftarrow o_t * tanh(C_t)$ ▷ output of lstm cell
38: $z_t \leftarrow ReLU(W_z h_t + b_z)$ ▷ dense layer
39: $prediction \leftarrow softmax(W_{sft} z_t + b_{sft})$
40: $Z_i \leftarrow Z_i \cup z_t$
41: **return** $Z_i$

---

model should be robust to person variance but it is very difficult to come up with a generalized model from the behavior of a limited number of individuals To this end, we perform person-independent experiments to emulate unseen conditions. Our train/test splits of the datasets are completely disjoint with respect to speakers.

While testing, our models have to classify emotions and sentiments from utterances by speakers they have never seen before.

**IEMOCAP:** The IEMOCAP contains the acts of 10 speakers in a two way conversation segmented into utterances. The database contains the following categorical labels: anger, happiness, sadness, neutral, excitement, frustration, fear, surprise, and other, but we take only the first four so as to compare with the state of the art (Rozgic et al., 2012b) and other authors. Videos by the first 8 speakers are considered in the train set. The train/test split details are provided in table 3.

**MOSI:** The MOSI dataset is a dataset rich in sentimental expressions where 93 persons review topics in English. It contains *positive and negative* classes as its sentiment labels. The train/validation set comprises of the first 62 individuals in the dataset.

**MOUD:** This dataset contains product review videos provided by around 55 persons. The reviews are in Spanish (we use Google Translate API[1] to get the english transcripts). The utterances are labeled to be either *positive, negative or neutral*. However, we drop the *neutral* label to maintain consistency with previous work. The first 59 videos are considered in the train/val set.

Table 3 provides information regarding train/test split of all the datasets. In these splits it is ensured that 1) *No two utterances from the train and test splits belong to the same video.* 2) *The train/test splits have no speaker overlap.* This provides the speaker-independent setting.

Table 3 also provides cross dataset split details where the complete datasets of MOSI and MOUD are used for training and testing respectively. The proposed model being used on reviews from different languages allows us to analyze its robustness and generalizability.

| Dataset | Train | | Test | |
|---|---|---|---|---|
| | *uttrnce* | *video* | *uttrnce* | *video* |
| IEMOCAP | 4290 | 120 | 1208 | 31 |
| MOSI | 1447 | 62 | 752 | 31 |
| MOUD | 322 | 59 | 115 | 20 |
| MOSI → MOUD | 2199 | 93 | 437 | 79 |

Table 3: uttrnce: Utterance; Person-Independent Train/Test split details of each dataset ($\approx$ 70/30 % split). Note: X→Y represents train: X and test: Y; Validation sets are extracted from the shuffled train sets using 80/20 % train/val ratio.

It should be noted that the datasets' individual configuration and splits are same throughout all the experiments (i.e., context-independent unimodal feature extraction, LSTM-based context-

---

[1] http://translate.google.com

dependent unimodal and multimodal feature extraction and classification).

## 4.2 Performance of Different Models and Comparisons

In this section, we present unimodal and multimodal sentiment analysis performance of different LSTM network variants as explained in Section 3.2.3 and comparison with the state of the art.

**Hierarchical vs Non-hierarchical Fusion Framework -** As expected, trained contextual unimodal features help the hierarchical fusion framework to outperform the non-hierarchical framework. Table 4 demonstrates this by comparing both hierarchical and non-hierarchical framework using the bc-LSTM network. Due to this fact, we provide all further analysis and results using the hierarchical framework. Non-hierarchical model outperforms the performance of the baseline *uni-SVM*. This further leads us to conclude that it is the context-sensitive learning paradigm which plays the key role in improving performance over the baseline.

**Comparison among Network Variants -** It is to be noted that both sc-LSTM and bc-LSTM perform quite well on the multimodal emotion recognition and sentiment analysis datasets. Since, bc-LSTM has access to both the preceding and following information of the utterance sequence, it performs consistently better on all the datasets over sc-LSTM.

The usefulness of the dense layer in improving the performance is prominent from the experimental results as shown in Table 4. The performance improvement is in the range of 0.3% to 1.5% on MOSI and MOUD datasets. On the IEMOCAP dataset, the performance improvement of bc-LSTM and sc-LSTM over h-LSTM is in the range of 1% to 5%.

**Comparison with the Baseline and state of the art -** Every LSTM network variant has outperformed the baseline *uni-SVM* on all the datasets by the margin of 2% to 5%(see Table 4). These results prove our initial hypothesis that modeling the contextual dependencies among utterances, which *uni-SVM* cannot do, improves the classification. The higher performance improvement on the IEMOCAP dataset indicates the necessity of modeling long-range dependencies among the utterances as continuous emotion recognition

is a multiclass sequential problem where a person doesnt frequently change emotions (Wöllmer et al., 2008).

We have implemented and compared with the current state-of-the-art approach proposed by Poria et al. (Poria et al., 2015). In their method, they extracted features from each modality and fed to a multiple kernel learning (MKL) classifier. However, they did not conduct the experiment in speaker-independent manner and also did not consider the contextual relation among the utterances. Experimental results in Table 5 shows that the proposed method has outperformed Poria et al. (Poria et al., 2015) by a significant margin. For the emotion recognition task, we have compared our method with the current state of the art (Rozgic et al., 2012b), who extracted features in a similar fashion to (Poria et al., 2015) did. However, for fusion they used SVM trees.

## 4.3 Importance of the Modalities

As expected, in all kinds of experiments, bimodal and trimodal models have outperformed unimodal models. Overall, audio modality has performed better than visual on all the datasets. On MOSI and IEMOCAP datasets, textual classifier achieves the best performance over other unimodal classifiers. On IEMOCAP dataset, the unimodal and multimodal classifiers obtained poor performance to classify *neutral* utterances. Textual modality, combined with non-textual modes boosts the performance in IEMOCAP by a large margin. However, the margin is less in the other datasets.

On the MOUD dataset, textual modality performs worse than audio modality due to the noise introduced in translating Spanish utterances to English. Using Spanish word vectors[2] in *text-CNN* results in an improvement of $10\%$ . Nonetheless, we report results using these translated utterances as opposed to utterances trained on Spanish word vectors, in order to make fair comparison with (Poria et al., 2015).

## 4.4 Generalization of the Models

To test the generalizability of the models, we have trained our framework on complete MOSI dataset and tested on MOUD dataset (Table 6). The performance was poor for audio and textual modality as the MOUD dataset is in Spanish while the model is trained on MOSI dataset which is in En-

---

[2]http://crscardellino.me/SBWCE

| Modality | MOSI hierarchical (%) | | | | non-hier (%) | MOUD hierarchical (%) | | | | non-hier (%) | IEMOCAP hierarchical (%) | | | | non-hier (%) |
|---|---|---|---|---|---|---|---|---|---|---|---|---|---|---|---|
| | uni-SVM | h-LSTM | sc-LSTM | bc-LSTM | | uni-SVM | h-LSTM | sc-LSTM | bc-LSTM | | uni-SVM | h-LSTM | sc-LSTM | bc-LSTM | |
| T | 75.5 | 77.4 | 77.6 | **78.1** | | 49.5 | 50.1 | 51.3 | **52.1** | | 65.5 | 68.9 | 71.4 | **73.6** | |
| V | 53.1 | 55.2 | 55.6 | **55.8** | | 46.3 | 48.0 | 48.2 | **48.5** | | 47.0 | 52.0 | 52.6 | **53.2** | |
| A | 58.5 | 59.6 | 59.9 | **60.3** | | 51.5 | 56.3 | 57.5 | **59.9** | | 52.9 | 54.4 | 55.2 | **57.1** | |
| T + V | 76.7 | 78.9 | 79.9 | **80.2** | 78.5 | 50.2 | 50.6 | 51.3 | **52.2** | 50.9 | 68.5 | 70.3 | 72.3 | **75.4** | 73.2 |
| T + A | 75.8 | 78.3 | 78.8 | **79.3** | 78.2 | 53.1 | 56.9 | 57.4 | **60.4** | 55.5 | 70.1 | 74.1 | 75.2 | **75.6** | 74.5 |
| V + A | 58.6 | 61.5 | 61.8 | **62.1** | 60.3 | 62.8 | 62.9 | 64.4 | **65.3** | 64.2 | 67.6 | 67.8 | 68.2 | **68.9** | 67.3 |
| T + V + A | 77.9 | 78.1 | 78.6 | **80.3** | 78.1 | 66.1 | 66.4 | 67.3 | **68.1** | 67.0 | 72.5 | 73.3 | 74.2 | **76.1** | 73.5 |

Table 4: Comparison of models mentioned in Section 3.2.3. The table reports the accuracy of classification. Note: non-hier ← Non-hierarchical bc-lstm. For remaining fusion hierarchical fusion framework is used (Section 3.3.2)

| Modality | Sentiment (%) | | Emotion on IEMOCAP (%) | | | |
|---|---|---|---|---|---|---|
| | MOSI | MOUD | *angry* | *happy* | *sad* | *neutral* |
| T | 78.12 | 52.17 | 76.07 | 78.97 | 76.23 | 67.44 |
| V | 55.80 | 48.58 | 53.15 | 58.15 | 55.49 | 51.26 |
| A | 60.31 | 59.99 | 58.37 | 60.45 | 61.35 | 52.31 |
| T + V | 80.22 | 52.23 | 77.24 | 78.99 | 78.35 | 68.15 |
| T + A | 79.33 | 60.39 | 77.15 | 79.10 | 78.10 | 69.14 |
| V + A | 62.17 | 65.36 | 68.21 | 71.97 | 70.35 | 62.37 |
| A + V + T | **80.30** | **68.11** | **77.98** | **79.31** | **78.30** | **69.92** |
| State-of-the-art | 73.55[1] | 63.25[1] | 73.10[2] | 72.40[2] | 61.90[2] | 58.10[2] |

[1] by (Poria et al., 2015), [2] by (Rozgic et al., 2012b)

Table 5: Accuracy % on textual (T), visual (V), audio (A) modality and comparison with the state of the art. For fusion, hierarchical fusion framework was used (Section 3.3.2)

| Modality | MOSI → MOUD | | | |
|---|---|---|---|---|
| | uni-SVM | h-LSTM | sc-LSTM | bc-LSTM |
| T | 46.5% | 46.5% | 46.6% | **46.9%** |
| V | 43.3% | 45.5% | 48.3% | **49.6%** |
| A | 42.9% | 46.0% | 46.4% | **47.2%** |
| T + V | 49.8% | 49.8% | 49.8% | **49.8%** |
| T + A | 50.4% | 50.9% | 51.1% | **51.3%** |
| V + A | 46.0% | 47.1% | 49.3% | **49.6%** |
| T + V + A | 51.1% | 52.2% | 52.5% | **52.7%** |

Table 6: Cross Dataset comparison. The table reports the accuracy of classification.

glish language. However, notably visual modality performs better than other two modalities in this experiment which signifies that in cross-lingual scenarios facial expressions carry more generalized, robust information than audio and textual modalities. We could not carry out the similar experiment for emotion recognition as no other utterance-level dataset apart from the IEMOCAP was available at the time of our experiments.

### 4.5 Qualitative Analysis

In some cases the predictions of the proposed method are wrong given the difficulty in recognizing the face and noisy audio signal in the utterances. Also, cases where the sentiment is very weak and non contextual, the proposed approach shows some bias towards its surrounding utter-

ances which further leads to wrong predictions. This can be solved by developing a context aware attention mechanism. In order to have a better understanding on roles of modalities for overall classification, we also have done some qualitative analysis. For example, this utterance - *"who doesn't have any presence or greatness at all."*, was classified as positive by the audio classifier ("doesn't" was spoken normally by the speaker, but "presence and greatness at all" was spoken with enthusiasm). However, textual modality caught the negation induced by "doesn't" and classified correctly. In another utterance "amazing special effects" as there was no jest of enthusiasm in speaker's voice and face audio-visual classifier failed to identify the positivity of this utterance. On the other textual classifier correctly detected the polarity as positive.

On the other hand, textual classifier classified this sentence - "that like to see comic book characters treated responsibly" as positive, possibly because of the presence of positive phrases such as "like to see", "responsibly". However, the high pitch of anger in the person's voice and the frowning face helps identify this to be a negative utterance.

### 5 Conclusion

Contextual relationship among the utterances is mostly ignored in the literature. In this paper, we developed a LSTM-based network to extract contextual features from the utterances of a video for multimodal sentiment analysis. The proposed method has outperformed the state of the art and showed significant performance improvement over the baseline. As a part of the future work, we plan to propose LSTM attention model to determine importance of the utterances and contribution of modalities in the sentiment classification.

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
