# Peer review of "Context-Dependent Sentiment Analysis in User-Generated Videos"

_ACL 2017 — decision unknown_

[Official Review · Reviewer 1 · rating 4 · confidence 3]
soundness 3 · originality 3 · clarity 5 · impact 3 · substance 4 · appropriateness 5 · meaningful comparison 3 · presentation format Oral Presentation

Dear Authors

thanks for replying to our review comments, which clarifies some detail
questions. I appreciate your promise to publish the code, which will be very
helpful to other researchers. 

Based on this, i increased my overall score to 4. 

Strengths:
- well-written
- extensive experiments
- good results

- Weaknesses:
- nothing ground-breaking, application of existing technologies
- code not available
- results are as could be expected

- General Discussion:
- why didn't you use established audio features such as MFCCs?

- Minor Details:
- L155 and other places: a LSTM -> an LSTM
- L160, L216 and other Places: why are there hyphens (-) after the text?
- L205: explanation of convolution is not clear
- Table1 should appear earlier, on page 2 already cited
- L263: is 3D-CNN a standard approach in video processing? alternatives?
- L375, 378: the ^ should probably positioned above the y
- L380: "to check overfitting" -> did you mean "to avoid"?
- L403, 408..: put names in " " or write them italic, to make it easier to
recognize them
- L420: a SVM -> an SVM
- L448: Output ... are -> wrong numerus, either "Outputs", or use "is" 
- L489: superflous whitespace after "layer"
- L516, 519: "concatenation" should not be in a new line
- L567: why don't you know the exact number of persons?
- L626: remove comma after Since
- L651: doesnt -> does not 
- L777: insert "hand, the" after other
- References: need some cleanup: L823 superflous whitespace, L831 Munich, L860
what is ACL(1)?, L888 superflous ), L894 Volume, L951 superflous new lines,
L956 indent Linguistics properly